# Amygdala Nuclei Atrophy in Cognitive Impairment and Dementia: Insights from High-Resolution Magnetic Resonance Imaging

**DOI:** 10.3390/medicina61010130

**Published:** 2025-01-15

**Authors:** Evija Peiseniece, Nauris Zdanovskis, Kristīne Šneidere, Andrejs Kostiks, Guntis Karelis, Ardis Platkājis, Ainārs Stepens

**Affiliations:** 1Department of Radiology, Riga Stradins University, LV-1007 Riga, Latvia; 008044@rsu.edu.lv (E.P.);; 2Department of Radiology, Riga East University Hospital, LV-1038 Riga, Latvia; 3Institute of Public Health, Riga Stradins University, LV-1007 Riga, Latviaainars.stepens@rsu.lv (A.S.); 4Department of Health Psychology and Paedagogy, Riga Stradins University, LV-1007 Riga, Latvia; 5Department of Neurology and Neurosurgery, Radiology, Riga East Clinical University Hospital, LV-1038 Riga, Latvia; andrejs.kostiks@rakus.lv (A.K.);; 6Department of Infectiology, Riga Stradins University, LV-1007 Riga, Latvia

**Keywords:** amygdala, amygdala atrophy, neuroradiology, structural magnetic resonance imaging, Montreal Cognitive Assessment, cognition, mild cognitive impairment, dementia

## Abstract

*Background and Objectives:* Cognitive impairment affects memory, reasoning, and problem-solving, with early detection being critical for effective management. The amygdala, a key structure in emotional processing and memory, may play a pivotal role in detecting cognitive decline. This study examines differences in amygdala nuclei volumes in patients with varying levels of cognitive performance to evaluate its potential as a biomarker. *Material and methods:* This cross-sectional study of 35 participants was conducted and classified into three groups: the normal (≥26), moderate (15–25), and low (≤14) cognitive performance groups based on the Montreal Cognitive Assessment (MoCA) scores. High-resolution magnetic resonance imaging at 3.0 T scanner was used to assess amygdala nuclei volumes. *Results:* Significant amygdala atrophy was observed in multiple amygdala nuclei across cognitive performance groups, with more pronounced changes in the low-performance group. The right hemisphere nuclei, including the lateral and basal nuclei, showed more significant differences, indicating their sensitivity to cognitive decline. *Conclusions:* This study highlights the potential of amygdala nuclei atrophy as a biomarker for cognitive impairment. Additional research with larger sample sizes and longitudinal designs is needed to confirm these findings and determine their diagnostic value.

## 1. Introduction

Cognitive impairment is a decline in cognitive function that affects essential abilities such as thinking, memory, reasoning, and problem-solving. These can range from mild impairments, such as mild memory loss, to more severe memory loss or even signs of dementia that significantly limit daily activities [1]. Compared to healthy individuals, patients with mild cognitive impairments suffer from more severe memory, concentration, and decision-making capacity loss, which, however, are not as severe as in the case of dementia [2].

The amygdala is an almond-shaped cluster of nuclei with a volume of roughly 1.1–1.7 cm^3^, located deep in the temporal lobes, about 1.5 cm from the lobe’s front and right under the anterior extremity of the parahippocampal gyrus. The amygdala is complex in structure. It comprises nine nuclei: the lateral nucleus, the basal nucleus, the accessory-basal nucleus, the anterior-amygdaloid area, the central nucleus, the medial nucleus, the cortical nucleus, and the para-laminar nucleus. All nuclei are divided into three groups: the ventrolateral or the basolateral group, the central group, and the dorsomedial or cortico-medial group [3,4].

Amygdala nuclei further subdivide into and form internuclear and intranuclear connections. Each nucleus is connected to different brain structures, such as neocortical and brainstem structures and other nuclei in the amygdala, and is involved in many neurophysiological processes. The amygdala manages information processing between the prefrontal-temporal association cortices and the hypothalamus [4,5,6].

As a part of the limbic system, the amygdala is crucial for main cognitive functions, such as emotional processing, memory formation, stress regulation, and neuroplasticity, demonstrating its importance in cognitive function [6,7].

In this study, we will compare the volume of amygdala atrophy measured by structural magnetic resonance imaging (MRI) in participants with mild cognitive impairment and dementia. We will investigate whether the microstructural changes in brain tissue, including the amygdala, detected by MRI could be an early biomarker for diagnosing these conditions [7,8]. We will compare the results obtained with the Montreal Cognitive Assessment (MoCA) data by assessing the severity of the study participants’ cognitive impairment [2,9,10].

Biomarkers are sensitive and specific for early neuropathological changes, thus identifying structural pathology caused by normal aging. They also help diagnose cognitive disorders, including cognitive impairment, which is critical in neurodegenerative disorders. This enables slowing cognitive decline, preventing irreversible neurodegenerative processes, and, as a result, improving the patient’s quality of life.

In their study, Heijer et al. (2009) revealed that elderly study participants (over 60 years of age) had hippocampal and amygdala MRI atrophy during a six-year follow-up, even without cognitive impairment and mild cognitive impairment [11].

Padullo et al. (2023) revealed in their study that there is a significant relationship between the atrophy of the right amygdala nuclei and the development of cognitive impairment. The study showed isolated amygdala volume reduction on MRI, without hippocampal or cerebral cortex atrophy [4].

Various studies have shown that individuals with mild cognitive impairment and Alzheimer’s disease exhibit significant atrophy in the amygdala. Poulin et al. (2011) established that reductions in amygdala volume correlate with deficits in memory and emotional processing in mild cognitive impairment, which predicts progression to Alzheimer’s disease. Emotional dysregulation is one of the hallmarks of dementia associated with amygdala dysfunction [1]. Balthazar et al. (2014) demonstrate the amygdala’s role in the manifestations of dementia, which include behavioral and psychological disturbances such as apathy and anxiety [12]. Pennanen et al. (2004) indicated that combining the diagnosis of amygdala atrophy with changes in the hippocampus enhances the accuracy of early Alzheimer’s disease detection [13].

Based on these findings, this study aims to investigate the relationship between amygdala atrophy and cognitive impairment by utilizing high-resolution MRI to measure the volume of the amygdala nuclei. By examining participants with varying levels of cognitive impairment, this study seeks to identify specific patterns of atrophy that may serve as early biomarkers of neurodegenerative conditions. Understanding these structural changes and their role in cognitive decline could lead to earlier diagnosis, personalized intervention, and, consequently, improved quality of life.

## 2. Materials and Methods

This cross-sectional study included 35 participants, who were divided into three groups according to the scores of the Montreal Cognitive Assessment (MoCA) scale [10,14,15]:Normal cognitive performance group (NPG)—study participants with scores ≥ 26;Moderate cognitive performance group (MPG)—study participants with scores ≤ 25 and ≥15;Low cognitive performance group (LPG)—study participants with scores ≤14.

There were 7 study participants in the NPG group (mean age 70.143, SD 6.362; the youngest participant was 61, the oldest 77; mean MoCA score 28.000, SD 1.155; lowest score 27; highest score 30).

There were 16 study participants in the MPG group (mean age 71.813, SD 6.483; youngest participant was 62, oldest 85; mean MoCA score was 21.563, SD 2.966; lowest score was 15, highest score was 25).

There were 12 participants in the LPG group (mean age 74.917, SD 10.535, the youngest participant was 62, oldest 96, mean MoCA score was 8.583, SD 3.872, the lowest score was 4, the highest score was 14). Table 1 shows the study participants’ demographic data (age), gender, and MoCA scores between the three groups.

A Chi-squared test on gender was performed, revealing no statistically significant differences between groups (X = 3.911, *p* = 0.141). Similarly, a Kruskal-Wallis test did not reveal age differences between groups (H (2) = 0.353, *p* = 0.838), but revealed statistically significant differences between MoCA scores and groups (H (2) = 29.260, *p* < 0.001). Spearman’s rho test discovered a statistically nonsignificant, negative, and weak correlation between age and MoCA score (r = −0.170, *p* = 0.328), as shown in Figure 1.

### 2.1. Selection of Study Participants

Participants included in our study were referred to a neurologist to evaluate the patient’s subjective complaints of cognitive impairment or suspected cognitive impairment based on a primary physician assessment. All the participants in this study had completed higher education.

Participants were excluded from our study if they had clinically significant neurological and psychiatric conditions, such as a history of tumors, major strokes, intracerebral lobar hemorrhages, vascular malformations, Parkinson’s disease, multiple sclerosis, major depression, or schizophrenia, as well as substance abuse, alcohol abuse, and other psychiatric conditions. The study also excluded participants with clinically verified vascular diseases and neurodegenerative and psychiatric diseases. Magnetic resonance imaging (MRI) revealed no clinically significant pathological findings in the study participants.

### 2.2. Magnetic Resonance Imaging Data Acquisition and Analysis

All the study participants underwent structural magnetic resonance imaging (MRI) using a SIGNA Architect 3.0 Tesla MRI scanner in a university hospital. T1, T2, MPRAGE, and 3D T1 (3D-T1 GRE) were used for MRI. ADNI (The Alzheimer’s Disease Neuroimaging Initiative) MRI protocol for biomarkers in the early diagnosis of Alzheimer’s Disease includes 3DT1, 3D FLAIR, T2 GRE, and DTI.

### 2.3. Amygdala Segmentation

Cortical reconstruction was performed by using FreeSurfer 7.2.0 image analysis software. It is documented and can be freely downloaded online: https://surfer.nmr.mgh.harvard.edu/fswiki/FreeSurferMethodsCitation (accessed on 1 September 2024). The technical details of these procedures are described in prior publications [16,17,18,19,20,21,22,23,24,25,26,27,28,29,30].

We used an automatic atlas created by an atlas-building algorithm based on Bayesian inference. This atlas, integrated into FreeSurfer, can automatically segment nine amygdala nuclei from a standard-resolution structural magnetic resonance image (see Figure 2). The atlas is publicly available in two datasets: ADNI and ABIDE. ADNI is documented and can be available for download online (https://adni.loni.usc.edu/data-samples/adni-data/ accessed on 1 September 2024).

Images were processed with FreeSurfer 7.2.0. The script “segmentHA_T1.sh” then calculated the amygdala’s segmentation into nuclei and quantified their volumes, considering the brain’s possible total volume. The amygdala was segmented into nine nuclei for the right and left hemispheres [31,32,33].

### 2.4. Statistical Analysis

The JASP statistical program (JASP 0.19.1) was used (created by Eric-Jan Wagenmakers, University of Amsterdam, Amsterdam, The Netherlands) when performing the study’s statistical analysis [34].

The study’s analysis consisted of descriptive statistics, the Chi-square test, Pearson’s correlation, Kruskal-Wallis, and Dunn’s post-hoc test with additional Bonferroni and Holm corrections.

Descriptive statistics was used to analyze the variables (age, gender, MoCA scores) and the differences between the three groups (NPG, MPG, and LPG). The Chi-Square test was used to analyze differences in categorical variables. We used Pearson’s correlation coefficient to find a significant relationship between two variables.

A Kruskal-Wallis test was used to determine statistically significant differences between the variable and three groups (NPG, MPG, and LPG). If the Kruskal-Wallis test revealed any statistically significant difference, the Dunn-Bonferroni post-hoc test was subsequently performed.

The Dunn-Bonferroni post-hoc test compares multiple pairs of means in a data group. To account for multiple comparisons and minimize the risk of type I errors, we applied post-hoc corrections, including Bonferroni corrections, after detecting significant results in nonparametric tests. This conservative strategy was chosen to ensure the robustness and reliability of the findings, given the study’s small sample size and exploratory nature. These statistical methods were employed to identify significant differences among cognitive performance groups while maintaining confidence in the results.

## 3. Results

### 3.1. Descriptive Statistics for Each Amygdala Nuclei on the Left and Right Hemispheres in Three Groups (NPG, MPG, and LPG) 

While conducting descriptive statistics for the amygdala nuclei of both cerebral hemispheres across all three cognitive performance groups (NPG, MPG, LPG), we observed a decrease in the mean values of all nuclei from NPG to LPG. This observation indicates potential structural or functional differences in the amygdala nuclei among the various groups. The LPG group exhibits the highest standard deviation (SD), showing greater variability in structural differences within the amygdala nuclei compared to the other groups. The results are presented in Table 2 and Table 3.

### 3.2. Statistical Differences Between the Three Groups (NPG, MPG, and LPG)

The Kruskal-Wall test was performed to determine statistically significant differences between the three groups (NPG, MPG, and LPG) for the amygdala nucleus of each hemisphere.

We did not find a statistically significant difference in the amygdala nuclei:Medial nucleus in the left hemisphere H (2) = 2.489, *p* = 0.288;Cortical nucleus in the right hemisphere H (2) = 5.634, *p* = 0.060;Para-laminar nucleus in the right hemisphere H (2) = 4.144, *p* = 0.126.

The Kruskal-Wallis revealed statistically significant differences between the three groups (NPG, MPG, and LPG) and volumes in subsequent nuclei in the left hemisphere amygdala:ALateral nucleus (H (2) = 6.614, *p* = 0.037);BBasal nucleus (H (2) = 9.935, *p* = 0.007);CAccessory-basal-nucleus (H (2) = 11.721, *p* = 0.003);DAnterior-amygdaloid-area (H (2) = 7.251, *p* = 0.027);ECentral nucleus (H (2) = 6.427, *p* = 0.040);FCortical nucleus (H (2) = 6.428, *p* = 0.040);GCortico-amygdaloid-transition area (H (2) = 8.606, *p* = 0.014);HPara-laminar nucleus (H (2) = 7.778, *p* = 0.020).

The Kruskal-Wallis test revealed statistically significant differences between the three groups (NPG, MPG, and LPG) and volumes in subsequent nuclei in the right hemisphere amygdala:ALateral nucleus in the right hemisphere (H (2) = 10.873, *p* = 0.004);BBasal nucleus in the right hemisphere (H (2) = 10.389, *p* = 0.006);CAccessory-basal-nucleus in the right hemisphere (H (2) = 11.304, *p* = 0.004);DAnterior-amygdaloid-area in the right hemisphere (H (2) = 6.918, *p* = 0.031);ECentral nucleus in the right hemisphere (H (2) = 13.669, *p* = 0.001);FMedial nucleus in the right hemisphere (H (2) = 8.151, *p* = 0.017);GCortico-amygdaloid-transition area (H (2) = 7.439 *p* = 0.024).

The raincloud plots visualize the statistically significant distribution in the left hemisphere’s (see Figure 3) and in the right hemisphere’s (see Figure 4) amygdala nuclei volumes (mm^3^) across the NPG, MPG, and LPG.

Dunn’s post-hoc test was performed to identify specific group differences. Significant differences are statistically highlighted in bold.

Dunn’s post-hoc test revealed statistically significant differences between LPG and MPG (z = −2.039, *p* = 0.041) and between LPG and NPG (z = −2.328, *p* = 0.020). Still, after the Bonferroni correction, there were no statistically significant differences (see Table 4).

Dunn’s post-hoc test revealed statistically significant differences between LPG and MPG (z = −2.231, *p* = 0.026) and between LPG and NPG (z = −3.002, *p* = 0.003). However, after the Bonferroni correction, there were no statistically significant differences between LPG and MPG, and the statistical difference between LPG and NPG was maintained (see Table 5).

Dunn’s post-hoc test was performed and revealed statistically significant differences between LPG and MPG (z = −2.130, *p* = 0.033) and between LPG and NPG (z = −3.359, *p* < 0.001). Still, after the Bonferroni correction, there were no statistically significant differences between LPG and MPG. After the Bonferroni correction, the statistical difference between LPG and NPG was maintained (see Table 6).

Dunn’s post-hoc test revealed statistically significant differences between LPG and MPG (z = −2.273, *p* = 0.023) and between LPG and NPG (z = −2.326, *p* = 0.020). However, after the Bonferroni correction, there were no statistically significant differences (see Table 7).

Dunn’s post-hoc test revealed statistically significant differences between LPG and NPG (z = −2.509, *p* = 0.012). After the Bonferroni correction, the statistical difference between LPG and NPG was maintained (see Table 8).

Dunn’s post-hoc test revealed statistically significant differences between LPG and NPG (z = −2.472, *p* = 0.013). After the Bonferroni correction, the statistical difference between LPG and NPG was maintained (see Table 9).

Dunn’s post-hoc test revealed statistically significant differences between LPG and NPG (z = −2.922, *p* = 0.003). After the Bonferroni correction, the statistical difference between LPG and NPG was maintained (see Table 10).

Dunn’s post-hoc test revealed statistically significant differences between LPG and NPG (z = −2.709, *p* = 0.007). After the Bonferroni correction, the statistical difference between LPG and NPG was maintained (see Table 11). 

Dunn’s post-hoc test revealed statistically significant differences between LPG and MPG (z = −2.540, *p* = 0.011) and between LPG and NPG (z = −3.034, *p* = 0.007). After the Bonferroni correction, the statistical differences between these groups were maintained (see Table 12). 

Dunn’s post-hoc test revealed statistically significant differences between LPG and MPG (z = −2.635, *p* = 0.008) and between LPG and NPG (z = −2.858, *p* = 0.004). After the Bonferroni correction, the statistical differences between these groups were maintained (see Table 13). 

Dunn’s post-hoc test revealed statistically significant differences between LPG and MPG (z = −2.231, *p* = 0.026) and between LPG and NPG (z = −3.259, *p* = 0.001). Still, after the Bonferroni correction, there were no statistically significant differences between LPG and MPG. After the Bonferroni correction, the statistical difference between LPG and NPG was maintained (see Table 14).

Dunn’s post-hoc test revealed statistically significant differences between LPG and MPG (z = −2.511, *p* = 0.012). After the Bonferroni correction, the statistical difference between LPG and NPG was maintained (see Table 15).

Dunn’s post-hoc test was performed and revealed statistically significant differences between LPG and MPG (z = −2.257, *p* = 0.024) and between LPG and NPG (z = −3.637, *p* < 0.001). Still, after the Bonferroni correction, there were no statistically significant differences between LPG and MPG. After the Bonferroni correction, the statistical difference between LPG and NPG was maintained (see Table 16).

Dunn’s post-hoc test revealed statistically significant differences between LPG and MPG (z = −2.204, *p* = 0.028) and between LPG and NPG (z = −2.624, *p* = 0.009). Still, after the Bonferroni correction, there were no statistically significant differences between LPG and MPG. After the Bonferroni correction, the statistical difference between LPG and NPG was maintained (see Table 17).

Dunn’s post-hoc test revealed statistically significant differences between LPG and MPG (z = −2.140, *p* = 0.032) and between LPG and NPG (z = −2.484, *p* = 0.039). However, after the Bonferroni correction, there were no statistically significant differences between LPG and MPG, and the statistical difference between LPG and NPG was maintained (see Table 18).

The study revealed statistically significant differences in the LPG-NPG groups in six nuclei of the left hemisphere of the amygdala: basal, accessory-basal, central, cortical, cortico-amygdaloid-transition area, and para-laminar, and in six nuclei of the right hemisphere of the amygdala: lateral, basal, accessory-basal, central, medial, and cortico-amygdaloid-transition area.

In addition, the study also revealed statistically significant differences between the LPG-MPG groups in the four nuclei of the right hemisphere of the amygdala: lateral, basal, anterior-amygdaloid area, and central.

Further, the study revealed statistically significant differences between the two groups, LPG-NPG and LPG-MPG, in the three nuclei of the right hemisphere of the amygdala: the lateral, basal, and anterior-amygdaloid areas.

## 4. Discussion

Nowadays, scientists aim to find effective quantitative biomarkers to diagnose early cognitive disorders. These biomarkers could detect cognitive impairment before the manifestation of clinical signs and show its development dynamics. Accordingly, patients could receive early personalized treatment that would reduce the progression of cognitive impairment and improve their quality of life.

Guan et al. (2017) described the heterogeneity of mild cognitive impairment and the degree of risk for developing dementia in their study. The research team also noted that mild cognitive impairment does not necessarily lead to dementia. Reversal conditions have been shown to delay cognitive improvement. Some studies have found that mild cognitive impairment with memory impairment as the predominant cognitive function is more likely to progress to dementia compared to cognitive impairment without memory loss [35,36,37,38].

Our study supports the validity of our research objective: amygdala atrophy could become a potential early biomarker that could help detect pathology and facilitate early treatment initiation. Many studies confirm that brain gray matter atrophy is one of the main biomarkers of Alzheimer’s disease [8,9,37]. Previous studies have found statistically significant atrophy of several brain regions, with the most pronounced reductions recorded in hippocampal and amygdala volumes [9,39,40].

This study aimed to reveal the relationship between structural changes in the amygdala of both cerebral hemispheres and mild or severe cognitive impairment. We investigated the morphological changes in the amygdala nuclei of each cerebral hemisphere among three groups with different cognitive impairments (NPG—normal cognitive performance group, MPG—moderate cognitive performance group, and LPG—low cognitive performance group). Our study found statistically significant differences in the amygdala nuclei of the right and left cerebral hemispheres. Qu et al. (2024) confirmed in their study that morphological changes in the amygdala nuclei may be more significant and sensitive than changes in the total volume of the amygdala [41].

Our study revealed that six different amygdala nuclei in each cerebral hemisphere showed statistically significant differences between the two study groups: LPG and NPG (in the amygdala nuclei of the right hemisphere—basal, accessory basal, central, cortical, paralaminar nuclei, and the cortico-amygdaloid transition area, and the amygdala nuclei of the left hemisphere—lateral, basal, accessory basal, central, medial, and the cortico-amygdaloid transition area).

The medial and central nuclei belong to the central group of the amygdala nuclei. A decrease in the volume of these nuclei may affect cognitive processes related to autonomic and behavioral responses to emotional stimuli associated with fear and anxiety [1,42,43,44,45]. This group of amygdala nuclei is interconnected with the visceral sensory and autonomic nuclei located in the brainstem and is involved in respiratory and cardiovascular functions [46,47].

In their study, Punzi et al. (2024) found that the basal, accessory basal, central, and lateral nuclei of the amygdala were the brain microstructures that showed statistically significant volume changes in patients with moderate cognitive impairment [44,48].

Jakob J. et al. (2022) found that right hemisphere amygdala atrophy plays a statistically significant role in the etiology of major depressive disorder. Seven amygdala nuclei were studied, of which only the central nucleus revealed a statistically significant difference with implications for the etiology of major depressive disorder [39,49,50].

The lateral and basal nuclei of the amygdala belong to the basolateral or deep group. When their volume decreases, cognitive processes related to sensory processing and information are affected [45,51,52,53].

In their study, Fox A.S. and Shackman A.J. (2019) highlight significant atrophy in the central nucleus of the amygdala and the nuclei of the basolateral group, which have distinct roles in cognitive and emotional processing. The central nucleus is crucial for autonomic and stress responses due to its connections with the brainstem and hypothalamus [43]. The basolateral group, which is involved in sensory integration and emotional regulation through connections with the prefrontal cortex and hippocampus, shows atrophy associated with deficits in emotional memory encoding and cognitive control [3,44]. The observed atrophy dominance in the right hemisphere aligns with theories of emotional processing asymmetry, where the right amygdala is more involved in negative emotional responses and appears more vulnerable to neurodegeneration [1]. These findings revealed that atrophy of the amygdala nuclei serves as a biomarker for dementia and underscored the need to explore their role within neural networks to understand the progression of cognitive and emotional processing deficits.

The cortical nucleus of the amygdala belongs to the corticomedial or dorsomedial group. As the volume of the cortical nucleus decreases, cognitive processes related to hunger and changes in eating habits are affected [44]. Researchers used the radial atrophy mapping technique to reconstruct the amygdala’s 3D surface, and a statistical analysis was performed on patients with Alzheimer’s disease and a control group. The results showed a statistically significant loss of amygdala volume in the cortical, lateral, basolateral, and medial nuclei in the Alzheimer’s disease group compared to the control group [45].

The paralaminar nucleus of the amygdala is located close to the Basolateral group. When its volume decreases, cognitive processes related to modulating emotional responses to social behavior and sensory and emotional information are affected [40,52].

The cortico-amygdaloid transition area of the amygdala is located near the medial nucleus and plays a crucial role in integrating sensory and emotional information. If the volume of this area decreases, cognitive disorders related to higher intellectual processes can arise, including cognitive and learning difficulties, emotional and cognitive balance regulation disorders, challenges in social interaction, and difficulties in decision-making under stress and fear [40,41,54,55].

Our study revealed statistically significant volume changes in four amygdala nuclei (lateral, basal, central, and the anterior amygdaloid area) of the right hemisphere between the two study groups: LPG–NPG and LPG–MPG. This important finding achieves the main goal of our study, which is to identify statistically significant changes among participants’ amygdala volumes with mild and severe cognitive impairment. The lateral and basal nuclei of the amygdala in the right cerebral hemisphere revealed statistically significant differences between these two study groups. A decrease in the volume of these amygdala nuclei in the participants of the studied groups can affect cognitive processes related to the formation of emotions and memories, reactions to stress, somatic and autonomic reflexes, and changes in eating and drinking behavior [38,45,48]. A decrease in the volume of the anterior amygdaloid area of the amygdala in study participants may impact cognitive processes crucial for processing emotions, responding to fear, and consolidating emotional memories [48,56].

Our study revealed a more pronounced atrophy of the right hemispheric amygdala, confirmed by several subsequent studies. Evidence from prior research indicates that the right hemisphere is more vulnerable to neurodegenerative changes in dementia. Studies show that the amygdala in the right hemisphere is primarily involved in processing emotional stimuli, particularly negative emotions like fear and stress, as well as in automatic regulation. This functional specialization may make it more susceptible to pathological changes associated with cognitive decline.

The study by Paloma F. Vazquez-Jimenez et al. (2023) showed a statistically significant decrease in the volume of five amygdala nuclei in the right cerebral hemisphere that was detectable before the onset of clinical signs, showing a difference between participants without cognitive impairment and participants with mild cognitive impairment between the ages of 60 and 90 [57]. Like many others, our study found a statistically significant difference between the amygdala nuclei of the right hemisphere and the amygdala nuclei of the left hemisphere [4,36,39]. Studies by Poulin et al. (2011), Wu et al. (2020), and Vazquez-Jimenez et al. (2023) have shown greater amygdala atrophy in the right hemisphere in Alzheimer’s disease and mild cognitive impairment, particularly in the lateral and basal nuclei. These findings support theories of hemispheric asymmetry, which suggest that the right hemisphere is more involved in global affective processing and is more susceptible to stress-induced neurodegeneration. The preferential atrophy observed in the amygdala of the right hemisphere in our study underscores its potential role as an early biomarker for dementia, highlighting the need for further research into the mechanisms driving this asymmetry [1,36,58].

In contrast to our study, Huang et al. (2023) found that amygdala atrophy predominated in the left hemisphere compared to the right hemisphere in the semantic dementia group. However, in the amnestic dementia group, amygdala atrophy was more significant in the right hemisphere than in the left hemisphere [57].

De-Wei Wang et al. (2021) demonstrated a statistically significant reduction in amygdala volume among study participants with Alzheimer’s disease neuropsychiatric disorders compared to those without the disease. Additionally, they found a positive correlation with the reduction in the hippocampal volume in both hemispheres [9,45,59,60].

Kang X. et al. (2024) collected numerous MRI images to identify atrophies in parts of the brain. They examined these images alongside various genetic profiles to uncover the biological causes of severe cognitive impairment. The study concluded that the hippocampus and amygdala exhibited the most pronounced atrophy. Genome-wide analysis indicated a statistically significant increase in the concentration of 5-HT1A receptors in the amygdala, which impacts the activity of neurotransmitters that affect memory [9,37,39].

Images were processed using FreeSurfer 7.2.0, a robust and well-validated tool for segmenting amygdala nuclei. However, it has its limitations and potential biases. One limitation is that FreeSurfer 7.2.0 relies on population-specific data, which may not fully capture the variability in anatomical structures across diverse groups. Moreover, the resolution of T1-weighted MRI sequences, commonly used for segmentation, may not sufficiently resolve smaller structures like the amygdala nuclei, potentially leading to inaccuracies. In cases of neurodegeneration, the FreeSurfer atlas may not align well with the altered anatomy, resulting in mislabeling or misclassification of anatomical regions. Furthermore, adjustments to voxel intensity for defining nuclei boundaries can introduce artifacts and errors in segmentation. Lastly, the segmentation protocol employed by FreeSurfer may differ from those used in other studies, potentially affecting comparisons of amygdala nuclei volumes across research settings. These limitations should be considered when interpreting the results, especially in the neurodegenerative contexts conditions.

In this study, we applied post-hoc corrections, such as Bonferroni corrections, to address multiple comparisons and reduce the risk of type I errors. While this conservative method helps ensure the robustness of statistically significant findings, it may have lowered sensitivity to subtle or borderline differences, possibly overlooking meaningful trends. This limitation is especially relevant in exploratory studies with small sample sizes, where achieving statistical power can be challenging. Implementing multiple testing corrections highlights the inherent trade-off between controlling false positives and maintaining sensitivity to detect true effects. Although some borderline significant results may have been excluded, the findings presented here indicate high confidence in the observed differences. Future studies with larger cohorts and alternative statistical methods, including Bayesian approaches, could aid in confirming and further exploring these potential trends while balancing error control with sensitivity.

The early detection of amygdala atrophy as a biomarker for cognitive impairment offers potential opportunities for clinical applications. Cognitive interventions, such as memory training and emotional regulation programs, are designed based on early detection; however, further studies are required to confirm their effectiveness in populations with amygdala atrophy [61]. Similarly, pharmacological approaches, such as cholinesterase inhibitors, may benefit from earlier initiation if amygdala atrophy is identified as part of a diagnostic protocol [62]. Lifestyle interventions, including structured aerobic exercise and stress reduction strategies, show promise in promoting brain health and could enhance early biomarker detection. Dietary approaches, such as following a Mediterranean diet, have also been associated with cognitive benefits and may contribute to a multidisciplinary strategy [63]. While these interventions show promise, additional research is crucial to determine the clinical impact and cost-effectiveness of incorporating amygdala atrophy as a biomarker into standard diagnostic practice protocols.

Amygdala atrophy could become a sensitive and specific biomarker for early neuropathological changes. This could help diagnose cognitive impairments that are fundamental to neurodegenerative disorders. It would also slow down the deterioration of cognitive functions, prevent irreversible neurodegenerative processes, and, as a result, improve the patient’s quality of life.

A cross-sectional design of the studies restricts the ability to assess progression over time. Nevertheless, the observed differences in amygdala nuclei volumes offer valuable insights into their potential role in cognitive impairment. Future longitudinal studies could further confirm these findings.

## 5. Conclusions

The study revealed that the volumes of most amygdala nuclei were statistically significantly different among the three groups: NPG, MPG, and LPG. Dunn’s post-hoc tests indicated statistically significant differences when comparing the NPG and LPG groups. Additionally, the lateral, basal, and central nuclei, along with the anterior amygdaloid area of the right hemisphere, showed statistically significant differences between the NPG and LPG groups and the MPG and LPG groups. A limitation of our study was the small sample size of study participants.

The findings of this study suggested that the reduction in the volume of the amygdala nuclei could serve as a biomarker for diagnosing early cognitive functions. In the future, cross-sectional studies with larger samples are necessary to confirm that the reduction in amygdala volume is a specific indicator of cognitive disorders. Furthermore, longitudinal studies are needed to confirm these findings and determine their diagnostic value.

## Figures and Tables

**Figure 1 medicina-61-00130-f001:**
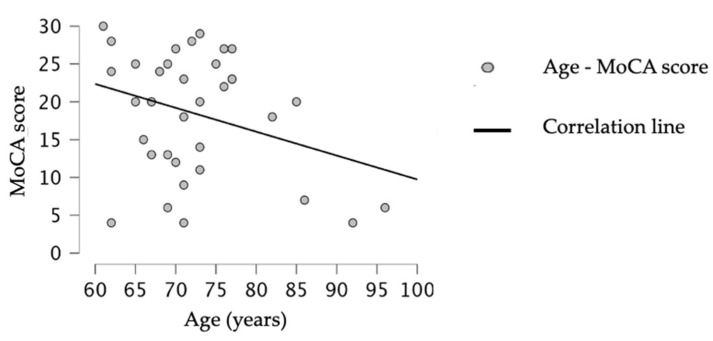
Spearman’s rho test (r = −0.170, *p* = 0.328) shows a correlation between age (years > 60) and MoCA score (0–30).

**Figure 2 medicina-61-00130-f002:**
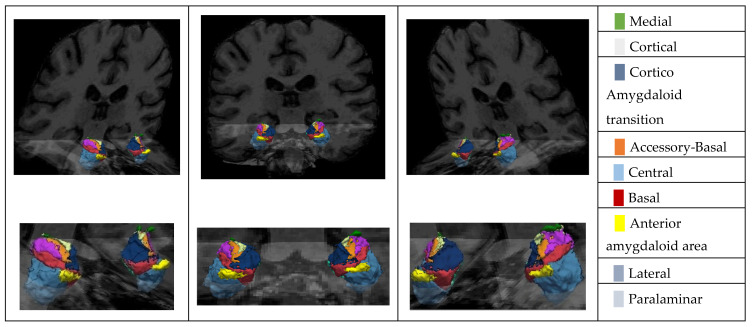
T1 MRI fusion with the 3D reconstruction of the Amygdala Nuclei.

**Figure 3 medicina-61-00130-f003:**
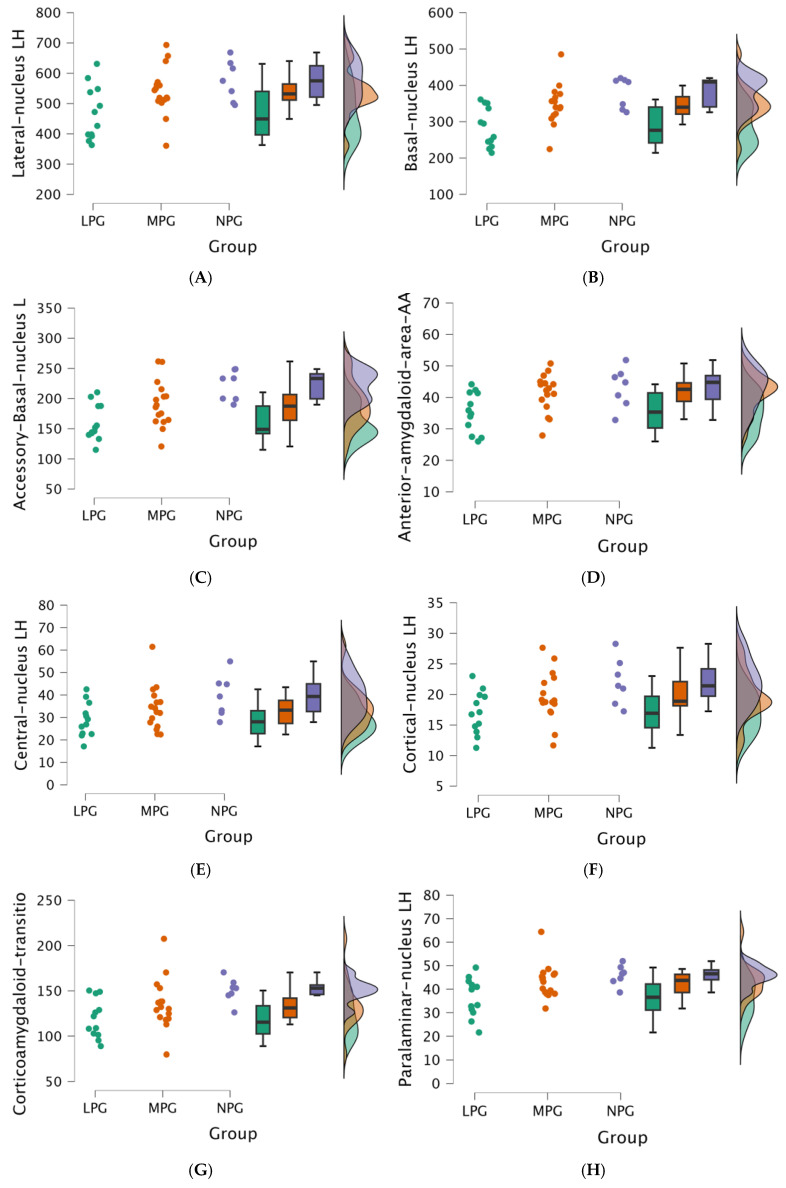
(**A**–**H**) Comparisons of the amygdala nuclei volumes (mm^3^) in the left hemisphere between three groups (LPG - green, MPG - moderate, and NPG – blue), including data distribution for each group. (**A**) Lateral nucleus LH (mm^3^). (**B**) Basal nucleus LH (mm^3^). (**C**) Accessory-basal-nucleus LH (mm^3^). (**D**) Anterior-amygdaloid-area LH (mm^3^). (**E**) Central nucleus LH (mm^3^). (**F**) Cortical Nucleus LH (mm^3^). (**G**) Cortico-amygdaloid- transition area LH (mm^3^). (**H**) Para-laminar nucleus LH (mm^3^).

**Figure 4 medicina-61-00130-f004:**
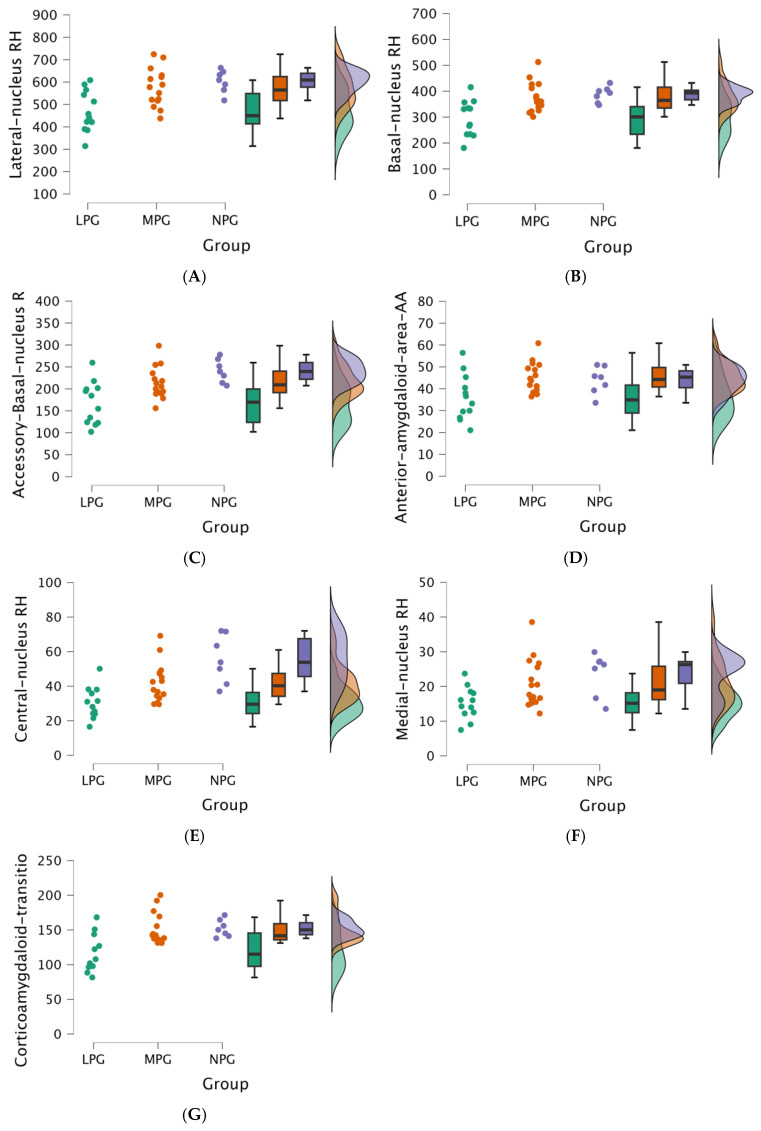
(**A**–**G**) Comparisons of the amygdala nuclei volumes (mm^3^) in the right hemisphere between three groups (LPG-green, MPG-orange, and NPG-blue), including data distribution for each group. (**A**) Lateral nucleus RH (mm^3^). (**B**) Basal nucleus RH (mm^3^). (**C**) Accessory-basal-nucleus RH (mm^3^). (**D**) Anterior-amygdaloid-area RH (mm^3^). (**E**) Central nucleus RH (mm^3^). (**F**) Medial nucleus RH (mm^3^). (**G**) Cortico-amygdaloid-transition area RH (mm^3^).

**Table 1 medicina-61-00130-t001:** Descriptive statistics for each group: normal cognitive performance group (NPG), moderate cognitive performance group (MPG), and kow cognitive performance group (LPG).

	Gender			Age			MoCA		
	LPG	MPG	NPG	LPG	MPG	NPG	LPG	MPG	NPG
Valid	12	16	7	12	16	7	12	16	7
Mean				74.917	71.813	70.143	8.583	21.563	28.000
Std. Deviation				10.535	6.483	6.362	3.872	2.966	1.155
Minimum				62.000	62.000	61.000	4.000	15.000	27.000
Maximum				96.000	85.000	77.000	14.000	25.000	30.000

**Table 2 medicina-61-00130-t002:** Descriptive statistics for each amygdala nucleus in the left hemisphere of the amygdala across three groups NPG, MPG, and LPG).

Amygdala Nuclei	Group	Mean	Std. Deviation	Minimum	Maximum
	NPG	575.716	66.753	495.050	668.370
**Lateral nucleus LH**	MPG	541.552	79.215	360.580	693.200
	LPG	468.132	89.677	362.870	630.880
	NPG	380.514	42.446	325.990	419.860
**Basal nucleus LH**	MPG	346.189	55.200	224.550	485.200
	LPG	284.690	54.535	214.250	361.050
	NPG	221.881	24.860	189.870	248.870
**Accessory-basal-nucleus LH**	MPG	190.771	38.138	120.630	261.560
	LPG	159.722	29.992	115.090	210.360
	NPG	43.149	6.391	32.810	51.840
**Anterior-amygdaloid-area LH**	MPG	41.353	6.044	27.880	50.770
	LPG	35.304	6.354	26.000	44.150
	NPG	39.633	9.379	27.920	54.970
**Central nucleus LH**	MPG	34.197	9.818	22.430	61.460
	LPG	28.947	7.612	17.120	42.540
	NPG	22.107	3.813	17.240	28.280
**Cortical nucleus LH**	MPG	19.616	4.107	11.660	27.640
	LPG	17.005	3.524	11.260	23.010
	NPG	150.657	13.660	126.210	170.420
**Cortico-amygdaloid-transition area LH**	MPG	135.531	28.015	79.910	207.530
	LPG	119.082	21.525	88.920	150.350
	NPG	45.941	4.287	38.640	51.930
**Paralaminar nucleus LH**	MPG	43.400	7.207	31.790	64.390
	LPG	36.325	8.319	21.610	49.210

**Table 3 medicina-61-00130-t003:** Descriptive statistics for each amygdala nucleus in the right hemisphere of the amygdala across three groups (NPG, MPG, and LPG).

Amygdala Nuclei	Group	Mean	Std. Deviation	Minimum	Maximum
	NPG	603.451	50.295	518.310	663.550
Lateral nucleus RH	MPG	572.451	83.325	437.690	724.270
	LPG	471.110	91.983	314.210	608.500
	NPG	387.784	30.047	346.920	432.080
Basal nucleus RH	MPG	376.715	56.997	301.370	512.900
	LPG	295.644	69.619	181.010	415.640
	NPG	241.213	26.495	207.340	277.940
Accessory-basal-nucleus RH	MPG	216.185	36.388	155.940	298.360
	LPG	167.826	48.700	102.110	259.660
	NPG	43.877	6.221	33.570	50.910
Anterior-amygdaloid-area RH	MPG	45.336	6.667	36.440	60.820
	LPG	36.053	10.456	21.040	56.420
	NPG	55.570	13.995	36.960	72.000
Central nucleus RH	MPG	41.989	11.225	29.510	69.150
	LPG	30.347	9.135	16.530	50.070
	NPG	23.690	6.124	13.530	29.910
Medial nucleus RH	MPG	20.964	6.911	12.210	38.550
	LPG	15.197	4.637	7.460	23.700
	NPG	152.384	12.359	138.000	171.430
Corticoamygdaloid-transition area RH	MPG	150.358	22.280	131.170	200.310
	LPG	121.209	30.436	81.540	168.340

**Table 4 medicina-61-00130-t004:** Dunn’s post-hoc comparison of the MoCA score and volume of the amygdala lateral nucleus in the left hemisphere.

Comparison	z	W_i_	W_j_	r_rb_	*p*	p_bonf_	p_holm_
LPG–MPG	−2.039	12.083	20.063	0.448	0.041	0.124	0.083
LPG–NPG	−2.328	12.083	23.429	0.667	0.020	0.060	0.060
MPG–NPG	−0.725	20.063	23.429	0.179	0.469	1.000	0.469

**Table 5 medicina-61-00130-t005:** Dunn’s post-hoc comparison of the MoCA score and volume of the amygdala basal nucleus in the left hemisphere.

Comparison	z	W_i_	W_j_	r_rb_	*p*	p_bonf_	p_holm_
LPG–MPG	−2.231	11.083	19.813	0.542	0.026	0.077	0.051
**LPG–NPG**	**−3.002**	**11.083**	**25.714**	**0.738**	**0.003**	**0.008**	**0.008**
MPG–NPG	−1.271	19.813	25.714	0.411	0.204	0.611	0.204

**Table 6 medicina-61-00130-t006:** Dunn’s post-hoc comparison of the MoCA score and volume of the amygdala accessory-basal nucleus in the left hemisphere.

Comparison	z	W_i_	W_j_	r_rb_	*p*	p_bonf_	p_holm_
LPG–MPG	−2.130	10.917	19.250	0.510	0.033	0.100	0.066
**LPG–NPG**	**−3.359**	**10.917**	**27.286**	**0.857**	**<0.001**	**0.002**	**0.002**
MPG–NPG	−1.731	19.250	27.286	0.518	0.084	0.251	0.084

**Table 7 medicina-61-00130-t007:** Dunn’s post-hoc comparison of the MoCA score and volume of the amygdala anterior-amygdaloid-area in the left hemisphere.

Comparison	z	W_i_	W_j_	r_rb_	*p*	p_bonf_	p_holm_
LPG–MPG	−2.273	11.667	20.563	0.521	0.023	0.069	0.060
LPG–NPG	−2.326	11.667	23.000	0.619	0.020	0.060	0.060
MPG–NPG	−0.525	20.563	23.000	0.161	0.600	1.000	0.600

**Table 8 medicina-61-00130-t008:** Dunn’s post-hoc comparison of the MoCA score and volume of the amygdala central nucleus in the left hemisphere.

Comparison	z	W_i_	W_j_	r_rb_	*p*	p_bonf_	p_holm_
LPG–MPG	−1.475	12.917	18.688	0.333	0.140	0.421	0.281
**LPG–NPG**	**−2.509**	**12.917**	**25.143**	**0.690**	**0.012**	**0.036**	**0.036**
MPG–NPG	−1.390	18.688	25.143	0.375	0.164	0.493	0.281

**Table 9 medicina-61-00130-t009:** Dunn’s post-hoc comparison of the MoCA score and volume of the amygdala cortical nucleus in the left hemisphere.

Comparison	z	W_i_	W_j_	r_rb_	*p*	p_bonf_	p_holm_
LPG–MPG	−1.634	12.667	19.063	0.365	0.102	0.306	0.204
**LPG–NPG**	**−2.472**	**12.667**	**24.714**	**0.690**	**0.013**	**0.040**	**0.040**
MPG–NPG	−1.217	19.063	24.714	0.321	0.224	0.671	0.224

**Table 10 medicina-61-00130-t010:** Dunn’s post-hoc comparison of the MoCA score and volume of the amygdala cortico-amygdaloid-transition area in the left hemisphere.

Comparison	z	W_i_	W_j_	r_rb_	*p*	p_bonf_	p_holm_
LPG–MPG	−1.576	12.333	18.500	0.375	0.115	0.345	0.164
**LPG–NPG**	**−2.922**	**12.333**	**26.571**	**0.762**	**0.003**	**0.010**	**0.010**
MPG–NPG	−1.738	18.500	26.571	0.500	0.082	0.247	0.164

**Table 11 medicina-61-00130-t011:** Dunn’s post-hoc comparison of the MoCA score and volume of the amygdala para-laminar nucleus in the left hemisphere.

Comparison	z	W_i_	W_j_	r_rb_	*p*	p_bonf_	p_holm_
LPG–MPG	−1.831	12.083	19.250	0.427	0.067	0.201	0.134
**LPG–NPG**	**−2.709**	**12.083**	**25.286**	**0.714**	**0.007**	**0.020**	**0.020**
MPG–NPG	−1.300	19.250	25.286	0.375	0.194	0.581	0.194

**Table 12 medicina-61-00130-t012:** Dunn’s post-hoc comparison of the MoCA score and volume of the amygdala lateral nucleus in the right hemisphere.

Comparison	z	W_i_	W_j_	r_rb_	*p*	p_bonf_	p_holm_
**LPG–MPG**	**−2.540**	**10.500**	**20.438**	**0.583**	**0.011**	**0.033**	**0.022**
**LPG–NPG**	**−3.034**	**10.500**	**25.286**	**0.810**	**0.002**	**0.007**	**0.007**
MPG–NPG	−1.044	20.438	25.286	0.304	0.296	0.889	0.296

**Table 13 medicina-61-00130-t013:** Dunn’s post-hoc comparison of the MoCA score and volume of the amygdala basal nucleus in the right hemisphere.

Comparison	z	W_i_	W_j_	r_rb_	*p*	p_bonf_	p_holm_
**LPG–MPG**	**−2.635**	**10.500**	**20.813**	**0.604**	**0.008**	**0.025**	**0.017**
**LPG–NPG**	**−2.858**	**10.500**	**24.429**	**0.762**	**0.004**	**0.013**	**0.013**
MPG–NPG	−0.779	20.813	24.429	0.232	0.436	1.000	0.436

**Table 14 medicina-61-00130-t014:** Dunn’s post-hoc comparison of the MoCA score and volume of the amygdala accessory-basal nucleus in the right hemisphere.

Comparison	z	W_i_	W_j_	r_rb_	*p*	p_bonf_	p_holm_
LPG–MPG	−2.231	10.833	19.563	0.531	0.026	0.077	0.051
**LPG–NPG**	**−3.259**	**10.833**	**26.714**	**0.833**	**0.001**	**0.003**	**0.003**
MPG–NPG	−1.540	19.563	26.714	0.464	0.124	0.371	0.124

**Table 15 medicina-61-00130-t015:** Dunn’s post-hoc comparison of the MoCA score and volume of the amygdala anterior-amygdaloid-area in the right hemisphere.

Comparison	z	W_i_	W_j_	r_rb_	*p*	p_bonf_	p_holm_
**LPG–MPG**	**−2.511**	**11.708**	**21.531**	**0.552**	**0.012**	**0.036**	**0.036**
LPG–NPG	−1.848	11.708	20.714	0.536	0.065	0.194	0.129
MPG–NPG	0.176	21.531	20.714	0.063	0.860	1.000	0.860

**Table 16 medicina-61-00130-t016:** Dunn’s post-hoc comparison of the MoCA score and volume of the amygdala central nucleus in the right hemisphere.

Comparison	z	W_i_	W_j_	r_rb_	*p*	p_bonf_	p_holm_
LPG–MPG	−2.257	10.417	19.250	0.563	0.024	0.072	0.048
**LPG–NPG**	**−3.637**	**10.417**	**28.143**	**0.881**	**<0.001**	**<0.001**	**<0.001**
MPG–NPG	−1.915	19.250	28.143	0.607	0.055	0.166	0.055

**Table 17 medicina-61-00130-t017:** Dunn’s post-hoc comparison of the MoCA score and volume of the amygdala medial nucleus in the right hemisphere.

Comparison	z	W_i_	W_j_	r_rb_	*p*	p_bonf_	p_holm_
LPG–MPG	−2.204	11.500	20.125	0.500	0.028	0.083	0.055
**LPG–NPG**	**−2.624**	**11.500**	**24.286**	**0.714**	**0.009**	**0.026**	**0.026**
MPG–NPG	−0.896	20.125	24.286	0.250	0.370	1.000	0.370

**Table 18 medicina-61-00130-t018:** Dunn’s post-hoc comparison of the MoCA score and volume of the amygdala cortico-amygdaloid-transition area in the right hemisphere.

Comparison	z	W_i_	W_j_	r_rb_	*p*	p_bonf_	p_holm_
LPG–MPG	−2.140	11.750	20.125	0.531	0.032	0.097	0.065
**LPG–NPG**	**−2.484**	**11.750**	**23.857**	**0.571**	**0.013**	**0.039**	**0.039**
MPG–NPG	−0.804	20.125	23.857	0.304	0.422	1.000	0.422

## Data Availability

The datasets presented in this article are not readily available, as they could contain potentially identifiable data. Requests to access the datasets should be directed to nauris.zdanovskis@rsu.lv.

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
