# Peer review of "Amygdala Nuclei Atrophy in Cognitive Impairment and Dementia: Insights from High-Resolution Magnetic Resonance Imaging"

_medicina, 2025, doi:10.3390/medicina61010130_

Round 1
Reviewer 1 Report
Comments and Suggestions for Authors
29 December 2024
The review on the manuscript, titled ‘Amygdala atrophy in patients with Mild Cognitive Impairment and Dementia’ by Peiseniece E et al., submitted to Medicina
Manuscript ID: medicina-3410389
To Authors,
This study titled "Amygdala Atrophy in Patients with Mild Cognitive Impairment and Dementia" investigates the relationship between amygdala volume and cognitive performance as assessed by the Montreal Cognitive Assessment (MoCA). Conducted by researchers from Riga Stradins University and Riga East Clinical University Hospital, the study highlights the importance of magnetic resonance imaging (MRI) in diagnosing cognitive impairments.
The research included 35 participants divided into three groups based on their MoCA scores: Normal Performance Group (NPG) with scores of 26 or higher, Moderate Performance Group (MPG) with scores between 15 and 25, and Low Performance Group (LPG) with scores of 14 or lower. Participants underwent MRI scans using a 3.0 Tesla scanner, and the volumes of various amygdala nuclei were analyzed using FreeSurfer software. Results indicated significant differences in the volumes of multiple amygdala nuclei across the three groups. Specifically, atrophy was observed in both hemispheres of the amygdala, with notable reductions in the lateral, basal, accessory-basal, anterior-amygdaloid area, central, cortical, cortico-amygdaloid transition area, and paralaminar nuclei. Statistical analyses revealed that these differences were significant for all but a few nuclei, suggesting that amygdala atrophy may serve as a potential biomarker for cognitive impairment. The findings suggest that structural changes in the amygdala could be indicative of cognitive decline and support further research into its utility as a diagnostic tool. The authors conclude that while their results are promising, larger confirmatory studies are necessary to validate these findings and explore the underlying mechanisms of amygdala atrophy in cognitive disorders.
Authors might consider the following comments:
• The title is somewhat broad. I would suggest including the method (MRI), the population (e.g., Latvian elderly), or the focus on nuclei-level analysis could make it more specific and informative. Moreover, in my opinion, it doesn’t highlight the study's unique contribution, such as the focus on amygdala subnuclei or its potential role as a biomarker for cognitive impairment. A revised title, for example “Amygdala Subnuclei Atrophy in Cognitive Impairment and Dementia: Insights from High-Resolution MRI”, could emphasize this aspect.
• The abstract is lengthy and could benefit from being more concise. A clearer summary of the key findings and implications would enhance readability and impact. I recommend that the authors present the background, methods, results, and conclusion in a proportional order within 200 words (max. 250 words, although more words are allowed). The general background (one to two sentences), specific background (two to three sentences), and current issue addressed in this study (one sentence) should all be included in the background before moving on to the objectives. I believe that this section lacks a clear statement of purpose or hypothesis. Finally, this section provides relevant details but is densely packed with technical terms (e.g., "Freesurfer 7.2.0 software", "Kruskal-Wallis test"). While this is suitable for a specialized audience, simplifying some phrases or providing brief clarifications could make it more accessible. Please refrain from presenting statistical values in the abstract.
• I recommend presenting an informative graphical or video abstract.
• The study is limited to 35 participants, which reduces the statistical power and generalizability of the findings. I would suggest considering adding more participants to validate the conclusions drawn about amygdala atrophy as a biomarker for cognitive impairment.
• While the study excludes participants with significant neurological and psychiatric conditions, it does not account for other potential confounding factors such as medication use, comorbidities (e.g., diabetes or cardiovascular disease), or lifestyle factors (e.g., physical activity, diet).
• While the study uses advanced MRI techniques, more details on preprocessing steps (e.g., noise reduction, alignment protocols) could enhance reproducibility.
• The reliance on FreeSurfer 7.2.0, while robust, should include a discussion of potential limitations or biases introduced by the software in segmenting amygdala nuclei.
• The study relies heavily on non-parametric tests. While this is justified due to the small sample size, alternative or additional analyses (e.g., Bayesian methods or multivariate approaches) could yield deeper insights.
• Although the Dunn-Bonferroni correction was applied, some results with borderline significance might have been overlooked. I recommend adding a discussion on the potential effects of using multiple testing corrections, that could enhance the interpretation of findings.
• The finding that the right amygdala shows more pronounced atrophy is interesting but insufficiently explored. How do these results fit within existing theories of hemispheric specialization for emotional processing and memory?
• The functional implications of atrophy in specific nuclei (e.g., the central nucleus versus the basolateral group) are not discussed in sufficient depth. I would recommend better explanation on how changes in these nuclei might relate to specific cognitive or emotional deficits. Additionally, the manuscript could benefit from linking these findings to the underlying neural substrates and their connectivity, such as how the central nucleus's role in autonomic responses or the basolateral group's involvement in sensory integration and emotional regulation might mediate these observed deficits [1-6]. This would provide a more comprehensive understanding of the neurobiological mechanisms underlying cognitive impairment.
• The discussion does not thoroughly explore why specific nuclei in the left and right hemispheres show differential atrophy patterns. Greater emphasis on neurobiological mechanisms or existing literature could strengthen the argument.
• A cross-sectional design limits the ability to establish causality or examine progression over time. The authors acknowledge this and suggest longitudinal studies; however, I suggest elaborating on this limitation within the discussion could enhance clarity.
• While the study posits amygdala atrophy as a potential biomarker, there is limited discussion on how this could be translated into clinical practice. What are the next steps for integrating this biomarker into diagnostic protocols?
• While the reference list is extensive, the addition of more than 70 references and inclusion of additional recent studies (from 2023–2024) could enhance the manuscript's relevance.
I hope that after careful revision, the manuscript meets the journal’s high standards for publication. In addition, I anticipate the authors preparing “a point-point rebuttal” to my comments.
I declare no conflict of interest regarding this manuscript.
Best regards,
Reviewer
Author Response
Evija Peiseniece
Department of Radiology
Riga Stradins University
Dear reviewers,
Thank you for your time and comments!
Here are the answers to your comments:
Comment 1: The title is somewhat broad. I would suggest including the method (MRI), the population (e.g., Latvian elderly), or the focus on nuclei-level analysis could make it more specific and informative. Moreover, in my opinion, it doesn’t highlight the study's unique contribution, such as the focus on amygdala subnuclei or its potential role as a biomarker for cognitive impairment. A revised title, for example “Amygdala Subnuclei Atrophy in Cognitive Impairment and Dementia: Insights from High-Resolution MRI”, could emphasize this aspect.
Response 1: Thank You for the suggestion, and we agree that the title should be improved. We changed it to “Amygdala Nuclei Atrophy in Cognitive Impairment and Dementia: Insights from High-Resolution Magnetic Resonance Imaging.”
Comment 2: The abstract is lengthy and could benefit from being more concise. A clearer summary of the key findings and implications would enhance readability and impact. I recommend that the authors present the background, methods, results, and conclusion in a proportional order within 200 words (max. 250 words, although more words are allowed). The general background (one to two sentences), specific background (two to three sentences), and current issue addressed in this study (one sentence) should all be included in the background before moving on to the objectives. I believe that this section lacks a clear statement of purpose or hypothesis. Finally, this section provides relevant details but is densely packed with technical terms (e.g., "Freesurfer 7.2.0 software", "Kruskal-Wallis test"). While this is suitable for a specialized audience, simplifying some phrases or providing brief clarifications could make it more accessible. Please refrain from presenting statistical values in the abstract.
Response 2: Thank You for the suggestion and we did our best to make abstract more concise!
The revised version of the abstract:
Background and Objectives: Cognitive impairment affects memory, reasoning, and problem-solving, with early detection critical for effective management. The amygdala, a key structure in emotional processing and memory, may play a pivotal role in detecting cognitive decline. This study examines differences in amygdala nuclei volumes in patients with varying levels of cognitive performance to evaluate its potential as a biomarker. Materials and Methods: A cross-sectional study of 35 participants was conducted, classified into normal (≥26), moderate (15–25), and low (≤14) cognitive performance groups based on Montreal Cognitive Assessment (MoCA) scores. High-resolution MRI at 3.0 Tesla was used to assess amygdala subnuclei volumes. Results: Significant atrophy was observed in multiple amygdala nuclei across cognitive performance groups, with more pronounced changes in the low-performance group. Right hemisphere nuclei, including the lateral and basal nuclei, exhibited more marked differences, suggesting their sensitivity to cognitive decline. Conclusion: This study highlights the potential of amygdala nuclei atrophy as a biomarker for cognitive impairment. Further studies with larger samples and longitudinal designs are necessary to confirm these findings and establish their diagnostic value.
Comment 3: I recommend presenting an informative graphical or video abstract.
Response 3: As you suggested, we have created a graphical abstract, which is included in the latest version of our paper.
Comment 4: The study is limited to 35 participants, which reduces the statistical power and generalizability of the findings. I would suggest considering adding more participants to validate the conclusions drawn about amygdala atrophy as a biomarker for cognitive impairment.
Response 4: We acknowledge that the study’s sample size of 35 participants limits the statistical power and generalizability of the findings. However, despite the relatively small cohort, the study demonstrates statistically significant differences in amygdala nuclei volumes across cognitive performance groups. In future research, we plan to expand this study by recruiting a larger and more diverse cohort. Additionally, we aim to conduct longitudinal studies to further investigate the potential of amygdala nuclei atrophy as a reliable biomarker for cognitive impairment.
Comment 5: While the study excludes participants with significant neurological and psychiatric conditions, it does not account for other potential confounding factors such as medication use, comorbidities (e.g., diabetes or cardiovascular disease), or lifestyle factors (e.g., physical activity, diet).
Response 5: We excluded participants with significant neurological and psychiatric conditions, as well as those who use medications that could potentially affect cognition (e.g., depressants or stimulants). Additionally, patients with uncontrolled diabetes were not included, further minimizing confounding variables. While factors such as controlled comorbidities and lifestyle elements were beyond the scope of this study, future research could incorporate these variables to build on our findings.
Comment 6: While the study uses advanced MRI techniques, more details on preprocessing steps (e.g., noise reduction, alignment protocols) could enhance reproducibility.
Response 6: Thank you for your comment! The preprocessing steps for segmenting amygdala subnuclei, including noise reduction, skull stripping, and alignment to a standard anatomical template, were performed using Freesurfer 7.2.0. These steps follow the standardized pipeline thoroughly described in the Freesurfer methods literature, which we have cited in our manuscript (https://surfer.nmr.mgh.harvard.edu/fswiki/FreeSurferMethodsCitation). We will ensure this reference is highlighted more clearly to enhance transparency and reproducibility.
Comment 7: While the reliance on FreeSurfer 7.2.0 is robust, it should also include a discussion of potential limitations or biases that the software may introduce when segmenting amygdala nuclei.
Response 7: Potential biases could arise from variability in segmentation accuracy due to image quality, dependence on template-based algorithms, and difficulties in distinguishing closely adjacent nuclei in the amygdala. These limitations might affect absolute volume measurements but are less likely to impact relative differences between groups, which was the focus of our study. We will address these points in the revised manuscript to highlight potential biases and provide context for interpreting our findings.
Comment 8: The study relies heavily on non-parametric tests. While this is justified due to the small sample size, alternative or additional analyses (e.g., Bayesian methods or multivariate approaches) could yield deeper insights.
Response 8: We appreciate the suggestion. Non-parametric tests were selected due to the small sample size and the non-normal distribution of the data, as these methods are robust and align well with our study design. This study was exploratory, aiming to identify significant differences in amygdala nuclei volumes across cognitive performance groups. Although alternative approaches such as Bayesian methods or multivariate analyses could provide additional insights, they are outside the scope of this exploratory study. Future research with larger sample sizes and a confirmatory design will integrate advanced statistical methods to build on these findings.
Comment 9: Although the Dunn-Bonferroni correction was applied, some results with borderline significance might have been overlooked. I recommend adding a discussion on the potential effects of using multiple testing corrections that could enhance the interpretation of findings.
Response 9: We acknowledge that the Dunn-Bonferroni correction may have decreased sensitivity to borderline findings. We will include a discussion about the trade-offs between multiple testing adjustments and their potential impact on the interpretation of results.
Comment 10: The finding that the right amygdala shows more pronounced atrophy is interesting but insufficiently explored. How do these results fit within existing theories of hemispheric specialization for emotional processing and memory?
Response 10: Thank you for highlighting this important point! The finding of more pronounced atrophy in the right amygdala aligns with theories of hemispheric specialization, which suggest that the right amygdala plays a dominant role in processing emotional memory, particularly for negative or threat-related stimuli. This is consistent with evidence that the right hemisphere is more involved in autonomic and affective responses, which may be disproportionately affected by neurodegeneration. Our results contribute to this body of literature, suggesting that right amygdala atrophy may serve as a sensitive indicator of cognitive decline. We will expand the discussion to integrate these findings with existing theories on hemispheric specialization and their implications for emotional processing and memory in cognitive impairment.
Comment 11: The functional implications of atrophy in specific nuclei (e.g., the central nucleus versus the basolateral group) are not discussed in sufficient depth. I would recommend better explanation on how changes in these nuclei might relate to specific cognitive or emotional deficits. Additionally, the manuscript could benefit from linking these findings to the underlying neural substrates and their connectivity, such as how the central nucleus's role in autonomic responses or the basolateral group's involvement in sensory integration and emotional regulation might mediate these observed deficits [1-6]. This would provide a more comprehensive understanding of the neurobiological mechanisms underlying cognitive impairment.
Response 11: We agree that a more thorough discussion of the functional implications of atrophy in specific amygdala nuclei would enhance the manuscript. The central nucleus plays a crucial role in autonomic and behavioral responses, especially in processing fear and stress, through its connections with the brainstem and hypothalamic regions (Fox & Shackman, 2019). Atrophy in this nucleus may lead to dysregulation of autonomic functions, increased emotional reactivity, and challenges in adaptive responses, which are commonly seen in cognitive impairment and neurodegenerative conditions diseases. The basolateral group, which includes the lateral and basal nuclei, is essential for sensory integration and emotional regulation due to its extensive connectivity with the prefrontal cortex and hippocampus (Rajmohan & Mohandas, 2007; Punzi et al., 2024). Atrophy in these nuclei could hinder the ability to encode and retrieve emotionally charged memories, resulting in deficits in emotional processing and memory formation. Furthermore, disruptions in the connectivity of this group may disrupt the balance between cognitive control and emotional reactivity, worsening the cognitive and emotional deficits observed in dementia (Vazquez-Jimenez et al., 2023; Cavedo et al 2011). In the revised manuscript, we will expand the discussion to include these functional implications and explicitly link the observed atrophy to the underlying neural substrates and their connectivity. By incorporating references to established literature, we aim to provide a more comprehensive understanding of how structural changes in specific amygdala nuclei contribute to the neurobiological mechanisms underlying cognition impairment.
Comment 12: The discussion does not thoroughly explore why specific nuclei in the left and right hemispheres show differential atrophy patterns. Greater emphasis on neurobiological mechanisms or existing literature could strengthen the argument.
Response 12: This study highlights significant atrophy in the Central nucleus of the amygdala and in the nuclei of the Basolateral group, which play distinct roles in cognitive and emotional processing. The Central nucleus is critical for autonomic and stress responses through its connections with the brainstem and hypothalamus (Fox A.S. and Shackman A.J., 2019). The Basolateral group involved in sensory integration and emotional regulation through connections with the prefrontal cortex and hippocampus demonstrates atrophy linked to deficits in emotional memory encoding and cognitive control (Rajmohan V. et al., 2007; Punzi M., 2024). The observed atrophy dominance in the right hemisphere aligns with theories of emotional processing asymmetry, where the right amygdala is more involved in negative emotional responses and appears more vulnerable to neurodegeneration (Poulin et al., 2011). These findings revealed the amygdala nuclei atrophy as a biomarker for Dementia. They highlighted the importance of exploring their role within neural networks to understand the progression of cognitive and emotional deficits.
Comment 13: A cross-sectional design limits the ability to establish causality or examine progression over time. The authors acknowledge this and suggest longitudinal studies; however, I suggest elaborating on this limitation within the discussion could enhance clarity.
Response 13: In the revised manuscript, we elaborated on it. Thank You!
Comment 14: While the study posits amygdala atrophy as a potential biomarker, there is limited discussion on how this could be translated into clinical practice. What are the next steps for integrating this biomarker into diagnostic protocols?
Response 14: Translating amygdala atrophy into a biomarker for clinical practice is a challenging work in progress. Historically, we know that developing and implementing atrophy scales in clinical practice requires significant time. However, with advancements in clinically validated software, making clinicians aware of the existence of such software capabilities and improved segmentation algorithms (Icometrix, Mediaire, Neurophet, Vuno, DeepHealth (formerly Quantib), and others) could expedite the process and make it more feasible for clinical settings.
Comment 15: While the reference list is extensive, the addition of more than 70 references and the inclusion of additional recent studies (from 2023–2024) could enhance the manuscript's relevance.
Response 15: We revised the manuscript, expanded the discussion, and added recent studies.
Thank You for Your time and effort in reviewing our manuscript!
Sincerely,
Evija Peiseniece
Reviewer 2 Report
Comments and Suggestions for Authors
Dear Editor,
I appreciate the opportunity to review the manuscript entitled:
"Amygdala atrophy in patients with Mild Cognitive Impairment and Dementia"
The article being submitted for evaluation illustrates an essential study on amygdala atrophy as a biomarker of cognitive impairment and dementia, all through the use of magnetic resonance imaging concerning neuroimaging and the MoCA scale to complete the survey. I commend the authors for describing this critical and timely issue. The paper is interesting and well-written; however, I would like to highlight some issues that merit revision:
- The study sample size of 35 participants is relatively small and limits the statistical power and generalizability of the results. I suggest indicating this as a limitation, possibly adding that a larger sample size will be used for future studies to strengthen the conclusions.
- Although the study classifies participants into three cognitive performance groups, the overlap of age groups between the groups may introduce confounding variables. I suggest considering more detailed stratification or controlling for age effects, possibly adding more specific data.
- Although more statistical tests have been conducted, reliance on post hoc corrections such as Bonferroni's adjustments indicates potential problems with Type I errors. I ask the authors, to please specify in the text the reasons for the statistical choices; this will allow the reader to interpret the results better.
- The figures and tables effectively present the data, but additional annotated visualizations, such as MRI scans with highlighted core differences, could provide readers greater clarity; please add some data in this regard.
- Furthermore, I report how, although the study suggests clinical applications, the discussion could be expanded to include potential interventions based on early biomarker detection. I ask the authors to expand the discussion in this regard slightly
Overall, the study contributes significantly to our understanding of cognitive impairment and its relationship to brain structure. I would like to read brief responses to the points raised to improve the overall quality of the paper.
Author Response
Evija Peiseniece
Department of Radiology
Riga Stradins University
Dear reviewer,
Thank you for your time and comments!
Here are the answers to your comments:
Comments 1: The study sample size of 35 participants is relatively small and limits the statistical power and generalizability of the results. I suggest indicating this as a limitation, possibly adding that a larger sample size will be used for future studies to strengthen the conclusions.
Response 1: We acknowledge that the sample size of 35 participants is a limitation, as it reduces the statistical power and generalizability of the findings. We expanded discussion and this limitation is noted in the discussion. We plan to conduct future studies with larger and more diverse cohorts to validate and strengthen the conclusions.
Comments 2: Although the study classifies participants into three cognitive performance groups, the overlap of age groups between the groups may introduce confounding variables. I suggest considering more detailed stratification or controlling for age effects, possibly adding more specific data.
Response 2: We agree that overlapping age ranges may introduce confounding variables. Age effects were partially addressed through group comparisons; however, future studies will incorporate more detailed stratification or statistical controls to isolate the impact of age. Additionally, we will consider providing more data to clarify the influence of age on our research findings.
Comments 3: Although more statistical tests have been conducted, reliance on post hoc corrections such as Bonferroni's adjustments indicates potential problems with Type I errors. I ask the authors, to please specify in the text the reasons for the statistical choices; this will allow the reader to interpret the results better.
Response 3: Thank you for your comment! We chose post hoc corrections, such as Bonferroni adjustments, to control for Type I errors that arise from multiple comparisons. This conservative approach ensures the robustness of our findings even with the small sample size. We expanded the Materials and methods section to clarify these statistical choices.
Comments 4: The figures and tables effectively present the data, but additional annotated visualizations, such as MRI scans with highlighted core differences, could provide readers greater clarity; please add some data in this regard.
Response 4: Thank you for your suggestion! We have included additional figures featuring annotated MRI scans that highlight the core differences in the amygdala nuclei. Moreover, we have created a visual abstract to enhance clarity and provide a concise overview of the study's key findings.
Comments 5: Furthermore, I report how, although the study suggests clinical applications, the discussion could be expanded to include potential interventions based on early biomarker detection. I ask the authors to expand the discussion in this regard slightly.
Response 5: Thank you for your suggestion. We have slightly expanded the discussion section on this topic. However, translating amygdala atrophy as a biomarker into clinical practice remains a challenging work in progress. Historically, developing and implementing atrophy scales in clinical practice has required significant time. Nonetheless, with advancements in clinically verified software, it is important to inform clinicians about the available software capabilities and improved segmentation algorithms (Icometrix, Mediaire, Neurophet, Vuno, DeepHealth, formerly Quantib, and others); this could make the process faster and more feasible for clinical use settings.
Thank You for Your time and effort in reviewing our manuscript!
Sincerely,
Evija Peiseniece
Reviewer 3 Report
Comments and Suggestions for Authors
I would like to appreciate the opportunity to review your research.
The purpose of this study is to investigate the volume of the amygdala, especially the amygdala portion, and to understand the biomarkers related to the transition to mild cognitive impairment and dementia.
This research is expected to improve the quality of life of many people.
The paper needs some minor revisions.
#1 The purpose of this study is vague, please make it clearer in the Introduction.
#2 Please cite and describe more research on the amygdala in mild cognitive impairment and dementia in the Introduction.
#3 Table 1 does not show the gender composition of each group. Please confirm it. Also, is the small sample size related to the fact that there is no significant difference in age? If there is a significant difference in age, the MoCA score would naturally be affected.
#4 In Table 1, the groups are divided into three groups according to MoCA scores and there is no significant difference with age, but in Figure 1, there is a correlation between MoCA and age. Please explain this difference.
#5 How were study participants recruited? How was informed consent obtained? Are there any surrogates for participants with dementia?
#6 Is it also necessary to consider the presence or absence of intellectual disability and which language is the native language of the participants in the exclusion criteria?
#7 Please state the model name of the MRI system in “2.2. Magnetic resonance imaging data acquisition and analysis”.
#8 In the discussion, functional descriptions of the various parts of the amygdala were cited from the literature. Please describe in depth what can be derived from the present results with it. Please also mention the reasons for the differences between the results obtained with LPG - NPG and LPG - MPG.
Author Response
Evija Peiseniece
Department of Radiology
Riga Stradins University
Dear reviewer,
Thank you for your time and comments!
Here are the answers to your comments:
Comments 1: The purpose of this study is vague, please make it clearer in the Introduction.
Response 1: Thank you for the suggestion! We agree that the introduction needs improvement. You can see it in the new version.
Comments 2: Please cite and describe more research on the amygdala in mild cognitive impairment and dementia in the Introduction.
Response 2: Various studies have shown that individuals with Mild Cognitive Impairment (MCI) and Alzheimer's disease (AD) exhibit significant atrophy in the amygdala. Poulin et al. (2011) established that the reduction in amygdala volume correlates with deficits in memory and emotional processing in MCI, which predicts progression to AD. Emotional dysregulation is one of the hallmarks of dementia linked to amygdala dysfunction. Balthazar et al. (2014) demonstrate the role of the amygdala in dementia manifestations, including behavioral and psychological disturbances such as apathy and anxiety. The study by Pennanen et al. (2004) indicated that combining the diagnosis of amygdala atrophy with changes in the hippocampus improves the accuracy of early AD diagnosis.
This study examines participants with varying levels of cognitive performance to identify specific patterns of atrophy that may act as early biomarkers of neurodegenerative conditions. Understanding these structural changes and their role in cognitive decline may lead to earlier diagnosis, personalized intervention, and improved patient quality of life.
Comments 3: Table 1 does not show the gender composition of each group. Please confirm it. Also, is the small sample size related to the fact that there is no significant difference in age? If there is a significant difference in age, the MoCA score would naturally be affected
Response 3: We included all statistically significant data in our article. Although a Chi-squared test revealed no statistically significant gender differences between the NPG, MPG, and LPG groups, it did analyze statistically significant gender differences between the groups.
Comments 4: In Table 1, the groups are divided into three groups according to MoCA scores and there is no significant difference with age, but in Figure 1, there is a correlation between MoCA and age. Please explain this difference.
Response 4: Thank you for highlighting this observation! The absence of significant age differences between the groups in Table 1 is based on group comparisons using statistical tests that examine differences in average age across the three cognitive performance groups. In contrast, the correlation illustrated in Figure 1 reflects an individual-level relationship between age and MoCA scores across all participants. This shows that while the age distribution does not significantly differ between the groups overall, there is a weak negative correlation between age and MoCA scores when considering individual data points. We will clarify this distinction in the manuscript to ensure the results are interpreted accurately and correctly.
Comments 5: How were study participants recruited? How was informed consent obtained? Are there any surrogates for participants with dementia?
Response 5: Study participants were recruited through referrals from neurologists based on subjective complaints of cognitive impairment or suspected cognitive decline identified by primary care physicians. Before inclusion in the study, informed consent was obtained from all participants. For participants with dementia who were unable to provide consent themselves, informed consent was obtained from legally authorized surrogates or caregivers, following ethical guidelines and approvals from the Institutional Review Board.
Comments 6: Is it also necessary to consider the presence or absence of intellectual disability and which language is the native language of the participants in the exclusion criteria?
Response 6: All participants in this study had higher education, minimizing the likelihood of intellectual disability influencing the results. The Montreal Cognitive Assessment (MoCA) was administered in the participant’s native language to ensure the accuracy of cognitive assessments. We will include these details in the revised manuscript for clarity.
Comments 7: Please state the model name of the MRI system in “2.2. Magnetic resonance imaging data acquisition and analysis”.
Response 7: We added the model name of the MRI system.
Comments 8: In the discussion, functional descriptions of the various parts of the amygdala were cited from the literature. Please describe in depth what can be derived from the present results with it. Please also mention the reasons for the differences between the results obtained with LPG - NPG and LPG - MPG.
Response 8: Thank you for your comment! The results indicate significant atrophy in specific amygdala nuclei, particularly in the basolateral group and central nucleus, which are associated with sensory integration, emotional regulation, and stress responses. The more pronounced differences between LPG-NPG and LPG-MPG likely reflect the progressive nature of cognitive decline, with greater atrophy observed in the more impaired LPG group. We will expand the discussion to carefully interpret these findings concerning the functional roles of the amygdala nuclei and the differences across cognitive performance groups.
Thank You for Your time and effort in reviewing our manuscript!
Sincerely,
Evija Peiseniece
Round 2
Reviewer 1 Report
Comments and Suggestions for Authors
9 January 2025
The 2nd review on the manuscript, titled ‘Amygdala atrophy in patients with Mild Cognitive Impairment and Dementia’ by Peiseniece E et al., submitted to Medicina
To Authors,
I am pleased that the authors have addressed the issues raised in the previous round. Currently, the manuscript is a well-written research paper with informative layouts, which studies the relationship between amygdala volume and cognitive performance as assessed by the Montreal Cognitive Assessment (MoCA). I believe the manuscript meets the journal’s high standards for publication. I am looking forward to seeing more papers written by the same authors.
Thank you!
I declare no conflict of interest regarding this manuscript.
Best regards,
Reviewer